# Reliability of Dynamic Shoulder Strength Test Battery Using Multi-Joint Isokinetic Device

**DOI:** 10.3390/s24113568

**Published:** 2024-06-01

**Authors:** Gustavo García-Buendía, Ángela Rodríguez-Perea, Ignacio Chirosa-Ríos, Luis Javier Chirosa-Ríos, Darío Martínez-García

**Affiliations:** 1Department of Physical Education and Sport, Faculty of Sports Sciences, University of Granada, 18011 Granada, Spain; gustavogb96@correo.ugr.es (G.G.-B.); ichirosa@ugr.es (I.C.-R.); dariomg@ugr.es (D.M.-G.); 2Strength & Conditioning Laboratory, CTS-642 Research Group, Department Physical Education and Sports, Faculty of Sport Sciences, University of Granada, 18001 Granada, Spain; 3Department Physical and Sport Education, University of León, 24071 León, Spain

**Keywords:** shoulder strength, isokinetic, athletic performance, injury, reproducibility, reliability

## Abstract

This study aimed to determine the absolute and relative reliability of concentric and eccentric flexion, extension, horizontal abduction, and adduction movements of the shoulder using a functional electromechanical dynamometer (FEMD). Forty-three active male university students (23.51 ± 4.72 years) were examined for concentric and eccentric strength of shoulder flexion, extension, horizontal abduction, and horizontal adduction with an isokinetic test at 0.80 m·s^−1^. Relative reliability was determined by intraclass correlation coefficients (ICCs) with 95% confidence intervals. Absolute reliability was quantified by the standard error of measurement (SEM) and coefficient of variation (CV). Reliability was very high to extremely high for all movements on concentric and eccentric strength measurements (ICC: 0.76–0.94, SEM: 0.63–6.57%, CV: 9.40–19.63%). The results of this study provide compelling evidence for the absolute and relative reliability of concentric and eccentric flexion, extension, horizontal abduction, and horizontal adduction shoulder isokinetic strength tests in asymptomatic adults. The mean concentric force was the most reliable strength value for all tests.

## 1. Introduction

Muscle strength is a key indicator of neuromuscular function and a crucial marker of physical fitness [1]. Strength assessment is a strong predictor of motor performance levels and identifying associated risks [2]. Assessing strength in the glenohumeral joint is complex due to its wide range of motion. This join relies on a combination of stabilizing and dynamic mechanisms to balance joint amplitude and stability [3]. 

The gold standard for shoulder strength evaluations often involves the use of isokinetic dynamometry due to its ability to provide precise, controlled, and reproducible measurements of muscle strength [4]. Isokinetic devices allow for the assessment of muscle performance at constant speeds, offering detailed insights into both concentric and eccentric muscle actions [5,6].

Despite the extensive research on shoulder strength, several gaps remain. While shoulder rotators have been extensively studied [7,8,9], the movements of flexion, extension, abduction, and adduction are equally crucial for functional activities but less frequently assessed in the literature. These movements are essential for various daily and athletic activities, including reaching, lifting, and throwing [10]. Assessing these movements provides a more comprehensive understanding of shoulder strength and can help identify potential weaknesses that might contribute to injury [1].

In recent years, a new kind of isokinetic device called multi-joint isokinetic machines have emerged [11]. These devices have shown validity [12] and reliability in the evaluation of muscle strength [13]. Moreover, these devices have already been used in previous studies to assess shoulder rotator strength [14,15]. Multi-join isokinetic devices provide good to excellent reliability in strength measurements, in addition to being user-friendly and more cost-effective than other similar devices [15]. 

Isokinetic tests usually show high intraclass correlation coefficients (ICCs), indicating excellent reliability [1]. However, it is crucial to note that reliability can vary depending on factors such as the speed of the test, the specific movements assessed, and the population being evaluated [11]. Conventional isokinetic devices report their data in angular velocities, ranging from slow (30–60 degrees per second) to high speeds (300–500 degrees per second) [16]. On the other hand, multi-joint isokinetic devices show their data in linear velocities ranging from 0.05 m·s^−1^ to 1.20 m·s^−1^ [7,11]. This introduces a challenge in the literature to compare the results of both types of devices [17].

Regarding the population evaluated to determine the reliability of these evaluations, although there are several studies with people with shoulder pathologies, athletes or elderly people, the main sample found is healthy young adults [18]. This population is often in optimal physical condition, making it an ideal population for establishing normative shoulder strength data for comparison with other populations [19,20].

Therefore, the primary objective of this study was to determine the absolute and relative reliability of a dynamic strength test battery in shoulder flexion, extension, horizontal abduction, and horizontal adduction, using a functional electromechanical dynamometer, in addition to comparing mean and peak force reliability in a seated position and determining the most reliable test condition. The research hypothesis is that this test will be a reliable method for the evaluation of concentric and eccentric strength in flexion, extension, horizontal abduction, and adduction of the shoulder; mean concentric force will be the most reliable condition; and more information about the glenohumeral joint can be obtained, both for injury prevention and readaptation, as well as for sports performance.

## 2. Materials and Methods

### 2.1. Participants

Forty-three active male university students (age: 23.51 ± 4.72 years, body mass: 80.13 ± 12.99 kg, height: 1.80 ± 0.13 m, and body mass index (BMI): 25.51 ± 2.88 kg/m^2^) participated in this study. These individuals had no prior experience with isokinetic or dynamometric devices. The study’s inclusion criteria were as follows: (i) absence of shoulder pain, with a maximum of 20% on the Shoulder Pain and Disability Index (SPADI); and (ii) no musculoskeletal injuries in the past six months. All participants were informed about the nature, objectives, and associated risks of the experimental procedure before providing their written consent to participate. The study protocol was approved by the University’s Biomedical Committee (no. 2884/CEIH/2022) and was conducted in accordance with the Declaration of Helsinki.

### 2.2. Experimental Design

A cross-sectional repeated measures design was employed to assess the strength of shoulder flexors, extensors, horizontal abductors, and horizontal adductors. Participants visited the laboratory on four separate days (with a minimum interval of 48 h) over two weeks, to complete two familiarization sessions, and two testing days. Each familiarization and testing day were identical, and participants completed isokinetic muscle strength evaluation for the four shoulder movements. Participants were instructed to maintain their level of physical activity throughout the two-week study period. All evaluations were conducted at the same time of day (±1 h) for each participant and under similar environmental conditions (~21 °C and ~60% humidity). The sequence of exercises was determined randomly.

### 2.3. Instruments

Shoulder flexion, extension, horizontal abduction, and horizontal adduction isokinetic strength evaluations were performed using an FEMD (Myoquality M1, Myoquality Solutions, Granada, Spain). FEMDs offer valid and reliable measurements of movement velocity and are appropriate devices for performing and evaluating natural movements in different planes [12]. The mechanical characteristics of this device include an accuracy of three millimeters for displacement, a variation of 100 g when determining a load, and a sampling frequency of 1000 Hz. Anthropometric data were measured using a BC-418 scale (Tanita Corporation, Tokyo, Japan) with a measurement error of 0.1 kg and a digital stadiometer HM200D (Charder Electronic, Taichung City, Taiwan) with a measurement error of 0.1 cm.

### 2.4. Procedures

The assessment and familiarization processes adhered to an identical procedure, with a warm-up period of 15 min in total. Initially, the general phase was on the cycle ergometer for five minutes. The subsequent five minutes were dedicated to joint mobility exercises and dynamic stretches using resistance bands. The final part, the specific phase of the warm-up, involved a single set of 10 repetitions each of shoulder flexion–extension, and shoulder horizontal abduction and adduction, executed in the position of the test. Following the warm-up, the participants assumed a seated position. The restraint system was adjusted according to the height of the participant and the force vector, with a variation of ±1 cm. Both the position and the 45-degree range of motion were determined using a base goniometer (Gymna hoofdzetel, Bilzen, Belgium). For the shoulder flexion test, the arm was positioned with a 45-degree flexion in the glenohumeral joint, and for the shoulder extension test, with a 90-degree flexion in the glenohumeral joint. For the horizontal abduction test, the position entailed a 90-degree flexion and a 45-degree adduction in the glenohumeral joint, and a 90-degree flexion and a 45-degree abduction in the glenohumeral joint for the shoulder horizontal adduction test. The elbow was kept extended, the forearm in 90-degree pronation, and the wrist aligned with the forearm in all tests. For the humeral–ulnar joint, the beginning of the scaphoid was used to place the force vector fixation and standardize the starting position (Figure 1).

At the onset of each testing session, participants were well rested, as they had a rest period of 5 min following the same warm-up and familiarization protocol before the commencement of the session. The test comprised two sets of 5 maximum consecutive repetitions of shoulder flexion, extension, horizontal abductors, and horizontal adductors at 0.80 m·s^−1^, within the previously established range of motion. The tests were randomly ordered using a computerized system. A rest period of three minutes was allowed between sets. The three highest repetitions of the mean force for both the concentric and eccentric contractions were recorded to calculate the mean dynamic force and the three highest repetitions of the peak force for both concentric and eccentric contractions were recorded to calculate the peak dynamic force for each participant. This measurement considered the average force of the total repetitions.

### 2.5. Statistical Analysis

Descriptive data are presented as mean ± standard deviation (SD). The normality of the data was assessed using the Shapiro–Wilk test, which is appropriate for small to moderate sample sizes due to its high power in detecting deviations from normality. Results from this test confirmed that the data were normally distributed (*p* > 0.05). 

Reliability was assessed by the intraclass correlation coefficient (ICC) with 95% confidence intervals. The ICC is a robust measure for test–retest reliability, particularly suitable for assessing the consistency of quantitative measurements.

Standard error of measurement (SEM) and coefficient of variation (CV) were also calculated to provide insights into absolute reliability. ICC values’ magnitude was classified through a qualitative scale, with values near 0.1 representing low reliability, 0.3 moderate, 0.5 high, 0.70 very high, and 0.9 extremely high [21]. SEM quantifies the precision of individual scores, while CV expresses the extent of variability in relation to the mean of the population. A CV value below 10% was considered indicative of acceptable reliability [22]. 

The level of agreement between force outcomes from two measures was also assessed using Bland–Altman plots and the calculation of systematic bias and its 95% limits of agreement (LoA = bias ± 1.96 SD) (Figure 2 and Appendix A). A customized spreadsheet was used to perform the reliability analysis [21], while JASP software (version 0.18.3. for MacOS, http://www.jasp-stats.org (accessed on 15 April 2024)) was used for the other analyses.

## 3. Results

The relative reliability for the flexion test was high or excellent in all conditions (ICC: 0.80 to 0.93). Absolute reliability ranged between 9.40% and 15.88%; between 9.40% and 11.39% in concentric contractions; and between 14.34% and 15.88% in eccentric contractions for CV and SEM, respectively (Table 1).

The relative reliability for the extension test was high or excellent in all conditions (ICC: 0.76 to 0.94). Absolute reliability ranged between 9.89% and 19.63%; between 9.89% and 11.31% in concentric contractions; and between 14.44% and 19.63% in eccentric contractions for CV and SEM, respectively.

The relative reliability for the abduction test was high or excellent in all conditions (ICC: 0.81 to 0.94). Absolute reliability ranged between 7.56% and 15.91%; between 7.56% and 10.65% in concentric contractions; and between 13.39% and 15.91% in eccentric contractions for CV and SEM, respectively.

For the adduction tests, the relative reliability was high or excellent in all conditions (ICC: 0.85 to 0.90). Absolute reliability ranged between 10.37% and 18.76%; between 10.37% and 17.32% in concentric contractions; and between 16.42% and 18.76% in eccentric contractions for CV and SEM, respectively. 

## 4. Discussion

The primary objective of this study was to determine the absolute and relative reliability of concentric and eccentric flexion, extension, horizontal abduction, and adduction movements of the shoulder using a functional electromechanical dynamometer (FEMD). To the best of our knowledge, this is the second study to evaluate the reliability of dynamic shoulder tests with multi-joint isokinetic devices.

The current research demonstrates very high to extremely high concentric and eccentric strength measurements for all movements (ICC: 0.76–0.94, SEM: 0.63–6.57%, CV: 9.40–19.63%). These results confirm that the evaluations conducted in this study provide a reliable method for assessing the flexion, extension, horizontal abduction, and adduction strength of the shoulder in asymptomatic adults. These findings have clinical relevance as shoulder strength can be a useful marker of shoulder joint function, both in terms of quality of life and athletic performance.

Lindström et al. (2003) reported ICC values between 0.86 and 0.90 in a shoulder flexion isokinetic test on healthy adults [23]. Those results are consistent with the current study’s ICC values. Although this study also tested in a seated position, the ROM (30° to 90° of shoulder flexion) and number of repetitions (3) were slightly different. Furthermore, this study only measured concentric contractions, and the angular velocities used (30°/s and 90°/s) are difficult to compare with linear velocities. On the other hand, there is no previous research reporting the ICC or SEM for a shoulder extension isokinetic test. Only Mayer et al. (1994) reported a 16.3% variability in the peak torque of a shoulder extension isokinetic test, but in a supine position, with belt stabilization and with a different range of motion, making it difficult to compare results with the current study [24]. 

Previous studies on the reliability of isokinetic tests of the shoulder abductor and adductor muscles have reported ICC values between 0.69 and 0.99 and CV values between 6% and 13% [24,25]. However, these movements were assessed in a vertical plane instead of a horizontal plane, and in a larger ROM. In the current study, similar reliability indices were found for both mean strength values and maximum strength values. Furthermore, this study observed less force developed in maximum voluntary concentric actions, increasing the variability in force development, and affecting reliability values, with eccentric variables being the most reliable evaluation conditions [19,26]. Given that the development of force in eccentric actions has a greater involuntary component, these actions could be considered maximum actions. A concentric action preceded by an eccentric one could benefit in terms of the involuntariness of maximum force development and improve recovery and injury prevention or enhance maximum force and power production in athletes [1,27,28].

The standard error of measurement (SEM) was included, as these values are crucial for the correct clinical interpretation of isokinetic dynamometry data [21,25]. The SEM is used to indicate the amount of measurement error in a single evaluation and to ascertain whether the difference in measurements between two individuals is real or due to measurement error [29]. This could be useful for calculating the sample size in future cross-sectional studies [25]. The SEM percentages in the current study suggest that the amount of measurement error for these measurement conditions varies between 0.63% and 6.57%. In addition, the SEM values obtained can be used to calculate the sample size in longitudinal studies to assess strength in asymptomatic patients, including shoulder isokinetic tests that aim to evaluate force variations [1].

Additionally, this study compares the absolute and relative reliability of the mean force and peak force. Mean force revealed a better absolute and relative reliability than peak force in concentric and eccentric phase for all movements. Furthermore, the mean concentric force presents the best absolute reliability (CV (%): 7.56–10.37, SEM (%): 0.63–1.28). On the other hand, eccentric peak forces showed the worst reliability values. Possible reasons are that an eccentric isokinetic test requires more familiarization [30], and peak force could overestimate the force produced due to greater variability, although it is most commonly used unit to assess shoulder strength [31].

A notable strength of this study is that it fills a gap in the literature on the assessment of strength for each shoulder movement [15]. Shoulder rotations are the most studied movements, but the rest of the movements had been less frequently assessed in the scientific literature. Therefore, this study provides a reliable new battery of tests to perform a complete evaluation of the shoulder in all its movements. It can be applied to both health and sporting environments with the aim of rehabilitation and injury prevention as well as to improve sport performance. Moreover, the use of FEMD requires an active stabilization of the proximal segments, in contrast to Velcro straps or belt stabilization of other isokinetic devices, to be able to apply the maximum possible force in a controlled natural movement [12]. This fact makes these strength test useful in the context of injury diagnoses, rehabilitation, and sport performance.

It is important to note some limitations of this study when evaluating the results. It is necessary to consider that the isokinetic test was performed with linear speeds, while most isokinetic tests are performed with angular speeds, which is a factor that makes it difficult to compare results with other studies due to the characteristics of the evaluation devices. Moreover, it was not possible to perform an inter-rater reliability analysis, so it is uncertain how this variable affects the evaluations. It is highly recommended that clinicians and trainers ensure the same evaluator is always with the same patient. Additionally, the study was conducted with asymptomatic active male university students, so the results cannot be applied to other populations such as patients with shoulder pain or sedentary individuals. Further study of these variables would be necessary to standardize the results to any type of population, including women, older adults, people with shoulder injuries people, or overhead athletes.

## 5. Conclusions

In conclusion, this study provides compelling evidence for the absolute and relative reliability of concentric and eccentric flexion, extension, horizontal abduction, and adduction movements of the shoulder using an FEMD. The results, demonstrating ICC and CV values within high to excellent ranges, confirm that the evaluations conducted in this study are reliable methods for assessing shoulder strength in asymptomatic adults. These findings align with previous research on the reliability of isokinetic tests of shoulder strength tests, further validating the use of these methods in clinical and sports science settings. Mean concentric force showed the most reliable strength values for all tests. The demonstrated reliability of these evaluations not only supports their use in assessing shoulder function in terms of quality of life and athletic performance but also suggests their potential utility in injury prevention and recovery. 

## Figures and Tables

**Figure 1 sensors-24-03568-f001:**
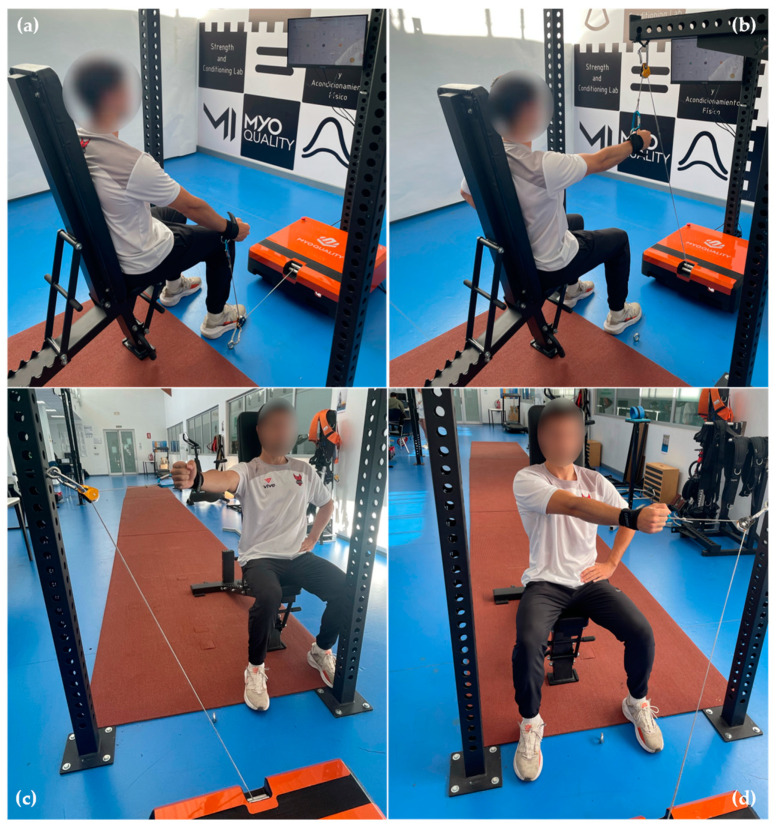
Initial position for the shoulder strength test battery. (**a**) Flexion; (**b**) extension; (**c**) horizontal abduction; (**d**) horizontal adduction.

**Figure 2 sensors-24-03568-f002:**
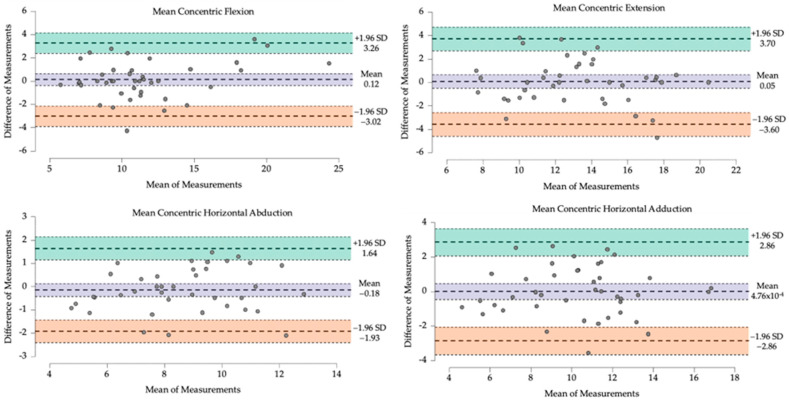
Bland Altman plots of test–retest for mean concentric forces.

**Table 1 sensors-24-03568-t001:** Reliability of the shoulder force isokinetic assessments using a functional electromechanical dynamometer.

			Mean ± SD		ICC	95% CI Lower–Upper	CV (%)	SEM (%)	ES	*p*
			Session 1	Session 2						
Flexion	Mean Force	Concentric	11.33 ± 4.09	10.90 ± 3.56	0.93	0.88–0.96	9.40	1.05	−0.07	0.257
		Eccentric	28.30 ± 9.76	28.05 ± 11.44	0.91	0.85–0.95	11.39	3.22	−0.02	0.694
	Peak Force	Concentric	24.29 ± 8.24	23.92 ± 9.08	0.80	0.67–0.89	15.88	3.79	−0.10	0.286
		Eccentric	40.91 ± 13.12	40.06 ± 15.04	0.84	0.72–0.91	14.34	5.82	−0.06	0.453
Extension	Mean Force	Concentric	12.92 ± 3.43	12.93 ± 3.73	0.87	0.78–0.93	9.89	1.28	0.01	0.944
		Eccentric	32.95 ± 14.63	32.30 ± 14.27	0.94	0.89–0.96	11.31	3.72	−0.05	0.394
	Peak Force	Concentric	26.97 ± 9.88	26.99 ± 11.28	0.76	0.60–0.86	19.63	5.32	−0.01	0.955
		Eccentric	45.66 ± 16.74	44.83 ± 16.86	0.85	0.75–0.92	14.44	6.57	−0.05	0.509
Abduction	Mean Force	Concentric	8.29 ± 2.27	8.39 ± 2.22	0.92	0.87–0.96	7.56	0.63	0.06	0.339
		Eccentric	22.64 ± 9.34	23.05 ± 9.86	0.94	0.89–0.96	10.65	2.47	0.05	0.391
	Peak Force	Concentric	15.79 ± 5.55	16.42 ± 6.45	0.81	0.69–0.89	15.91	2.63	0.15	0.104
		Eccentric	30.11 ± 11.29	30.98 ± 12.15	0.88	0.79–0.93	13.39	4.15	0.07	0.329
Adduction	Mean Force	Concentric	9.64 ± 3.31	9.32 ± 3.00	0.90	0.82–0.94	10.37	1.02	0.03	0.648
		Eccentric	25.72 ± 12.73	25.25 ± 11.71	0.87	0.78–0.93	17.32	4.46	−0.03	0.629
	Peak Force	Concentric	20.68 ± 10.53	19.49 ± 10.47	0.89	0.81–0.94	16.42	3.53	0.00	0.968
		Eccentric	33.68 ± 16.64	33.53 ± 16.00	0.85	0.75–0.91	18.76	6.44	−0.03	0.629

SD: Standard deviation; ICC: intraclass correlation coefficient; CI: confidence interval; CV: coefficient of variation; SEM: standard error of measurement; ES: effect size.

## Data Availability

Data can be made fully available upon request.

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
