# Peer review of "Reliability of Dynamic Shoulder Strength Test Battery Using Multi-Joint Isokinetic Device"

_sensors, 2024, doi:10.3390/s24113568_

Round 1

Reviewer 1 Report

Comments and Suggestions for Authors

General comment

This is an interesting study; however, several aspects should be improved before I endorse its publication. The most important aspect that needs special attention is the language, as the text will benefit from a language editor. Regarding the sections, the introductions needs major revision.

Specific comments

l.15: add information about the ‘adults’: what is the sex? who are they, e.g. students?

l.15: report age with no or one decimal

l.23-26: the conclusions should focus on the findings; this part is not about the significance or practical applications of the study.

l.67: The introduction till l.67 is pertinent should be deleted since it does not focus on the research problem. Instead I suggest that the authors focus on questions such as (a) what is the validity of this test (because without speaking about the validity of a test does not make sense to discuss about reliability)? (b) why the focus should be in a young population? (c) why a 0.80 m/s speed should be of interest?, (d) what is the interest to study these motions and not the rotations?, (d) what is the gap in the existing knowledge?, (e) what is known about the reliability of this test so far?

l.92: see l.15 comment

l.105: what the testing of each day included?

l.118: planes e?

l.142: describe the positions of elbow, forearm and wrist joint during the tests

Comments on the Quality of English Language

The most important aspect that needs special attention is the language, as the text will benefit from a language editor. 

Reviewer 2 Report

Comments and Suggestions for Authors

1. Improve statistical methods: In the section of statistical analysis, it is suggested to add testing methods and results of data normality, as well as reasons for choosing to use specific statistical methods.

2. Increased discussion of study limitations: The paper should discuss in detail the limitations of the study, including sample size, measurement errors, experimental design limitations, etc., and how these limitations may affect the interpretation and generalization of the study results.

3. Extended discussion section: In the discussion section, the differences and similarities between the results of this study and previous studies should be compared and compared, and the possible methodological reasons behind these differences should be explored. At the same time, the significance and application prospect of the research results for clinical practice, sports training and injury prevention should also be discussed.

Comments on the Quality of English Language

Moderate editing of English language required

Round 2

Reviewer 1 Report

Comments and Suggestions for Authors

All concerns have been addressed.

Reviewer 2 Report

Comments and Suggestions for Authors

The author has made the necessary modifications as requested.

Comments on the Quality of English Language

Minor editing of English language required